# An Old New Friend: Folliculo-Stellate Cells in Pituitary Neuroendocrine Tumors

**DOI:** 10.3390/cells14131019

**Published:** 2025-07-03

**Authors:** Valeria-Nicoleta Nastase, Iulia Florentina Burcea, Roxana Ioana Dumitriu-Stan, Amalia Raluca Ceausu, Flavia Zara, Catalina Poiana, Marius Raica

**Affiliations:** 1Department of Microscopic Morphology/Histology, “Victor Babes” University of Medicine and Pharmacy, 300041 Timisoara, Romania; ra.ceausu@umft.ro (A.R.C.); flavia.zara@umft.ro (F.Z.); raica@umft.ro (M.R.); 2Angiogenesis Research Centre, “Victor Babes” University of Medicine and Pharmacy, 300041 Timisoara, Romania; 3Pius Branzeu” County Emergency Hospital, 300041 Timisoara, Romania; 4“C. I. Parhon” National Institute of Endocrinology, 011863 Bucharest, Romania; iulia.burcea@umfcd.ro (I.F.B.); roxana-ioana.dumitriu@drd.umfcd.ro (R.I.D.-S.); endoparhon@gmail.com (C.P.); 5Endocrinology Department, “Carol Davila” University of Medicine and Pharmacy, 020021 Bucharest, Romania; 6Municipal Emergency Hospital, 300041 Timisoara, Romania

**Keywords:** PitNETs, GFAP, FS cells, blood vessels

## Abstract

Pituitary neuroendocrine tumors (PitNETs) represent a complex pathology based on numerous incompletely elucidated molecular mechanisms. Beyond tumor cells, analyzing the tumor microenvironment may help identify novel prognostic markers and therapies. A key component of this environment is the folliculo-stellate (FS) cell. We examined FS cells in 77 PitNETs obtained by transsphenoidal surgery, using glial fibrillary acidic protein (GFAP) as an immunohistochemical marker. Immunohistochemistry for anterior pituitary hormones and transcription factors was performed to accurately classify the tumors. Our study included 19 somatotroph, 16 mammosomatotroph, 5 plurihormonal PIT-1 positive, 7 corticotroph, 14 gonadotroph, 11 unusual plurihormonal, and 5 null cell PitNETs. FS cells were observed in 55 of the cases, distributed isolated, in small groups or diffuse networks. A considerable number of tumors immunopositive for more than one hormone (including associations between GH/PRL, but also unusual combinations like GH/ACTH) also contained FS cells (*p* < 0.01), suggesting their involvement in tumor lineages differentiation. In 27 tumors, GFAP-positive cells clustered in highly vascularized areas. Additionally, in 11 of these cases a direct interaction between endothelial cells and FS cells was noted, sustaining their potential role in tumor angiogenesis. Given their complexity, FS cells may be crucial for understanding tumorigenesis mechanisms.

## 1. Introduction

Pituitary neuroendocrine tumors (PitNETs) represent common lesions derived from anterior pituitary glandular cells and they may be incidentally identified in up to a fifth of the general population [1]. Since 2017, the classification of these tumors is based on adenohypophyseal cell lineages, established using pituitary hormones and their transcription factors, according to World Health Organization (WHO) [2]. Even if most of PitNETs have a slow progression and they are successfully treated using surgery or drugs, some possess an aggressive behavior and their clinical management is still difficult to perform [1,3]. In this context, it is mandatory to understand the mechanisms involved in tumorigenesis, in order to develop new targeted treatments. Through the study of other tumors, it is well known that, in addition to tumoral cells, the tumor microenvironment (TME) plays an important role in the process [4]. Despite this aspect, the TME in PitNETs is scarcely studied; it includes extracellular matrix, cytokines, blood and lymph vessels, fibroblasts, immune cells and folliculo-stellate (FS) cells.

FS cells are nonsecretory cells located in the anterior pituitary gland, described by having a star-like appearance and the ability to form follicles [5]. One of the most special characteristics of these cells is represented by their long cytoplasmic processes. Each of these long cytoplasmic processes surround neighboring endocrine cells. Also, they were observed to have a close relationship with blood vessels [6]. They were first identified using electron microscopy [7] and their recognition is easily made by their lack of cytoplasmic granules and by the presence of tiny follicles that they produce [8].

Using immunohistochemical techniques, FS cells were first characterized by S100-protein antibody, and later other additional substances specific for FS cells were mentioned, among them being glial fibrillary acidic protein (GFAP) [9], vimentin, and SOX10 [10].

FS cells regulate endocrine cells, controlling their microenvironment [11] by producing a multitude of paracrine factors. Among the first factors described were basic fibroblast growth factor and vascular endothelial growth factor (VEGF), both being potent stimulators of angiogenesis [12]. Also, it was observed that FS cells contain some lysosomes in the cytoplasm and perform phagocytotic activity [13]. Based on the association between FS cells and immature endocrine cells, the possibility that the FS cell is a type of stem cell with potential to differentiate into endocrine cells has also been mentioned [14].

Despite the fact that there are some studies focused on the presence of FCs cells in PitNETs [15], their entire role in the pathogenesis of these tumors and connections between them and tumors subtypes, evolution, and prognostic are poorly understood [16].

The current study is a retrospective observational analysis aiming to evaluate the characteristics and distribution of FS cells within PitNETs, in relation to their specific subtypes, as defined according to current immunohistochemical classification.

## 2. Materials and Methods

The current study included 77 patients diagnosed with PitNETs based on clinical data, biochemical assays, and imagistic evaluation. The study has the approval of the University of Medicine and Pharmacy “Victor Babes” Ethics Committee (Ethics Approval no.102/03.10.2022 rev 2025). All patients underwent transsphenoidal surgery in the Neurosurgery Clinic of “Bagdasar Arseni” Emergency Clinical Hospital (Bucharest, Romania), the Neurosurgery Clinic of “Colentina” Hospital (Bucharest, Romania), and the Neurosurgery Clinic of the Brain Institute, Monza Hospital (Bucharest, Romania), respectively. After sampling, tissue fixation was performed using 10% neutral buffered formalin, and all primary processing steps were completed, resulting in paraffin-embedded blocks that were subjected to morphological and immunohistochemical analysis at the Department of Histology and Angiogenesis Research Center, “Victor Babes” University of Medicine and Pharmacy, Romania. Histopathological diagnosis was performed following routine staining with hematoxylin and eosin (H&E) on 3 µm sections for each case. The quality of the specimens was confirmed using vimentin immunostaining (ETU, Leica, clone V9, RTU). Morphological staining was conducted using Leica Autostainer XL (Leica Biosystem Newcastle Ltd., Balliol Business Park West, Benton Lane, New Castle Upon Tyne NE 12 EW, United Kingdom). Microscopic evaluation was assessed with the Nikon Eclipse E 600 microscope (Nikon Corporation, Tokyo, Japan).

All cases included in the study were characterized according to the established diagnostic algorithms and criteria, which were based on the immunohistochemical expression of anterior pituitary hormones and their specific transcription factors. The characteristics of FS cells were evaluated in immunohistochemical slides assessing the expression of GFAP. The primary antibodies used can be found in Table 1. Bond Epitope Retrieval Solution 1 and 2 with pH values of 6 and 9, respectively, were used for unmasking (Leica Biosystems, Newcastle Ltd., Newcastle Upon Tyne NE 12 8EW, UK). Additionally, 3% hydrogen peroxide was used to block endogenous peroxidase for 5 min.

Statistical analysis was performed using DATAtab (DATAtab Team 2025. DATAtab: Online Statistics Calculator. DATAtab e.U. Graz, Austria. URL https://datatab.net) statistical analysis tool was used. Frequencies of the categorical variables (Histopathological features, PitNET classification, FS cell characteristics) were presented as number or percentage or both where relevant.

The Chi2 test was performed to verify the associations between FS cell presence, PitNET subtypes, and number of positive hormones.

## 3. Results

### 3.1. Histophatological Evaluation

Histopathological analysis (using H&E staining) established the definitive diagnosis (pituitary neuroendocrine tumor) and classified the tumors according to their growth pattern: 45.45% (*n* = 35) of cases presented a diffuse architectural pattern, 25.97% (*n* = 20) were alveolar tumors followed by 23.38% (*n* = 18) papillary tumors; the fewest of the tumors were described as having trabecular growth pattern, 5.19% (*n* = 4) (Figure 1).

### 3.2. Immunohistochemichal Analysis

Immunohistochemical analysis represented a crucial step in order to accurately classify PitNETs according to current guidelines. Following the evaluation of adenohypophyseal hormones (GH, PRL, TSH, ACTH, FSH, LH) and pituitary transcription factors, we described the following PitNETs subtypes: 19 somatotroph (immunopositive for GH, PIT1), 16 mammosomatotroph (immunopositive for GH, PRL, PIT1), 5 plurihormonal PIT-1 positive (positive for GH, TSH, and/or PRL, PIT1), 7 corticotroph (immunopositive for ACTH, TPIT), 14 gonadotroph (immunopositive for FSH, LH, or both and SF1), 11 unusual plurihormonal (immunopositive for different hormones and transcription factors), and 5 null cell tumors (immunonegative for all hormones and transcription factors). Among the tumors with unusual plurihormonal associations, we found 7 PitNETs positive for GH and ACTH (Figure 2), 3 PitNETs immunopositive for GH, LH and 1 PitNET immunopositive for GH, TSH, LH.

The immunopositivity for pituitary hormones and transcription factors is described in Table 2.

FS cells were evaluated by immunohistochemical techniques using GFAP. Thus, a distinct population of cells was identified, separate from those involved in hormone secretion. The immunohistochemical analysis revealed a positive cytoplasmic reaction in 55 of PitNETs (71.43%). There were observed star-shaped cells, characterized by numerous elongated extensions located among the tumoral cells. Three forms of distribution of these cells have been described: isolated (low density) at the tumoral level, organized in nests/groups with focal arrangement inside the tumoral mass (medium density), and respectively arranged in diffuse networks throughout the tumor (high density). Additionally, we observed the development of follicular structures in 17 cases: 5 somatotroph, 4 mammosomatotroph, 1 plurihormonal PIT 1 positive, 4 gonadotroph and 3 unusual plurihormonal PitNETs (Figure 3).

There was significant variability observed between the presence of FS cells and PitNETs subtypes. Positive immunohistochemical reaction for GFAP was identified in 9 somatotroph, 16 mammosomatotroph, 5 plurihormonal PIT1-1 positive, 5 corticotroph, 10 gonadotroph, 9 unusual plurihormonal, and one null cell PitNETs. An important number of tumors immunopositive for more than one hormone (including associations between: GH/PRL, GH/PRL/TSH, but also unusual combinations like GH/ACTH, GH/LH, GH/TSH/LH) contained FS cells (*p* < 0.01) also. A high density of GFAP-positive cells arranged in diffuse networks inside tumoral mass was observed in 20 of cases, while 19 cases presented small groups of FS cells. Isolated positive cells were found in 16 cases. The association between PitNETs subtypes and distribution forms of FS cells is described in Table 3.

In the current study we analyzed the interactions between FS cells, blood vessels, and endothelial cells. In 27 of PitNETs GFAP-positive cells were concentrated in richly vascularized tumoral areas (small groups of cells or diffuse networks). Moreover, in 11 PitNETs presenting diffuse networks of FS cells, a direct interaction between endothelial cells and GFAP-positive cells (Figure 4) was noted.

### 3.3. Statistical Analysis

A Chi2 test was performed between PitNET subtype and FS cells. At least one of the expected cell frequencies was less than 5. Therefore, the assumptions for the Chi2 test were not met.

A Chi2 test was performed between the FS cells and the number of positive hormones. All expected cell frequencies were greater than 5; thus, the assumptions for the Chi2 test were met. With the Chi2 test, Cramér’s V was used to calculate the effect strength. Cramér’s V = 0.43 indicated an effect strength between medium (0.3) and large (0.5). There was a statistically significant relationship between FS cells and the number of positive hormones, χ^2^ (1) = 14.07, *p* = <0.001. The Pearson contingency coefficients were calculated to measure the strength of the relationship between the two variables. A Pearson contingency coefficient of 0.56 suggests a moderate to strong association between the variables being analyzed in the Chi Squared test.

## 4. Discussion

Pituitary tumors are mostly benign, but sometimes they may have various behaviors: invasive, aggressive, and/or malignant with metastases. Since 2017 (according with World Health Organization), they are divided into seven morphofunctional types and three lineages: somatotroph, lactotroph, and thyrotroph (PIT1 lineage), corticotroph (TPIT lineage) or gonadotroph (SF1 lineage), null cell or immunonegative tumor and plurihormonal tumors. The current classification is based on advances in physiology, cell biology, and genetics [17]. The International Pituitary Pathology Club, a group formed by experienced scientists, endocrinologists, pathologists, and neurosurgeons, have proposed that pituitary adenomas be recognized as PitNETs [18].

According to the latest World Health Organization classification of PitNETs, a new subtype of tumor was mentioned: the mature plurihormonal Pit-1 lineage tumor, being similar with a mammosomatotroph adenoma (positive for GH and/or PRL), but in addition presenting immunopositivity also for TSH [19,20]. We identified five of these PitNETs, all of which exhibited FS cells with a distribution either isolated, in small groups, or diffuse networks. Also, all mammosomatotroph tumors included in the current study (16 cases) contained GFAP-positive cells. Somatotroph PitNETs (immunopositive only for GH) exhibited a lower density of FS cells compared to the previously mentioned tumors.

A study based on pituitary tumors tissue obtained from 286 patients with GH-secreting tumors also highlighted increased variability in FS cells distribution. Similar to our results, pituitary tumors expressing more than one adenohypophyseal hormone presented a higher density of FS cells [21].

In addition to the FS cells described in the literature as having neuroectodermal origin [22] and expressing markers such as GFAP, S100, Sox10 [22,23,24], a distinct subset of S100 positive cells, namely follicular cells, has also been described. These cells usually present immunoexpression for keratins CAM 5.2, AE1/AE3 and for CK 18, CK 19. It is suggested that these cells derive from endocrine cells, and they were described only in gonadotroph PitNETs. In approximately one quarter of these tumors, star-shaped keratin positive cells delimitate follicles, or are located along vascular channels. Our current study included 14 gonadotroph tumors and 10 of them presented GFAP-positive cells. In half of these cases, FS cells exhibited increased density, forming extensive intratumoral networks [6,16].

Follicular cells, the subset previously described, commonly organize in follicular structures filled with colloid that contains clusterin [25]. Clusterin represent a cellular and circulating glycosylated protein that is believed to be involved in both progression and regression of tumors [26]. Follicles delimited by follicular cells are usually localized around necrotic tumoral areas, suggesting their involvement in phagocytic actions. Some studies suggest that intracellular clusterin is associated with gonadotroph PitNETs tumoral restraint mediated through activation of p53/p21, p15, p16, but also through forkhead transcription factor FOXL2 involvement [27,28]. In contrast, in our study, follicular structures were identified only in four gonadotroph tumors, without describing a direct association between presence of follicles and PitNETs subtype. This aspect could suggest distinct categories of FS cells immunopositive for GFAP (used in current study) and S100 (used in anterior-mentioned studies).

An interesting aspect of our study is represented by the important number of unusual plurihormonal PitNETs, 11 cases (14%). Moreover, 9 of them also presented FS cells distributed mainly in focal groups or diffuse networks. These tumors express hormones belonging to more than one cell lineage and are considered a rare tumoral subtype. It is supposed that numerous factors like specific genes, transcription factors, interactions between different signaling pathways engage in the pathogenesis of these rare PitNETs. None of them is completely understood so far. Additionally, to the nomenclature included in 2022 WHO classification, the term “multilineage” tumor was proposed to better define this current pathology. These types of tumors defined by multiple cell lineage combinations are mentioned to have worse short-term prognoses with lower complete response rates. The most common subtype mentioned in the literature is represented by the PIT 1-SF1 multilineage tumor [29,30,31,32]. Among our cases, we identified four types of these PitNETs, and two of them expressed GFAP-positive cells.

Although multilineage PIT-1/TPIT are rarer tumors compared to PIT-1/SF1 positive PitNETs, our study included seven of this tumoral type. Also, GH and ACTH, the hormones related to these cell lineages, were expressed. There are reported only 21 cases of tumors co-expressing GH and ACTH [33]. A significant aspect of our research is that all PIT1/TPIT positive PitNETs exhibited FS cells. Based on this, FS cells involvement in the tumorigenesis process and differentiation of multiple tumor lineages may be suggested.

Another interesting aspect of our study is represented by the important number of tumors immunopositive for more than one hormone, 40 PitNETs, respectively (we included among these cases also the associations between GH/PRL and FSH/LH). Moreover, 36 of these cases (90%) contained GFAP immunopositive cells. This aspect may suggest a direct influence exerted by GFAP-positive cells on the tumoral cells selection and differentiation. Furthermore, the ability of these cells to transform into a hormone-secreting adenohypophyseal cell lineage could be mentioned. Sox2, a transcription factor considered to define a stem cell phenotype, was identified in a subset of FS cells included in GH-secreting PitNETs [34]. Through electronic microscopy, a few immature endocrine cells were identified in contact with FS cells, delimitating an incomplete follicle. In this context, the authors have suggested the involvement of these cells in endocrine cell differentiation [10]. The plasticity of FS cells and their ability to transform into other types of cells, like striated muscle cells, were also confirmed in vitro [35]. The growing interest in pituitary stem cells has underscored the diverse functions that FS cells play in the physiology of the pituitary gland. Understanding each of these roles and their interconnections may become crucial in developing new treatments for pituitary disorders [36].

Recent advances have highlighted the possibility that a subpopulation of stem-like cells may contribute to PitNETs pathogenesis, mirroring the established cancer stem cell (CSC) model established in malignant tumors. Although most of these tumors are benign and effectively managed with surgical or pharmacological treatments, approximately 35% display aggressive behavior, marked by elevated proliferative capacity, invasiveness, and a high likelihood of recurrence. This clinical variability emphasizes the necessity for a more comprehensive understanding of the molecular pathways involved in tumorigenesis process. Although the WHO classification integrates pituitary-specific transcription factors and emphasizes tumor cell characteristics, the underlying biological mechanisms driving tumor initiation and progression—particularly those associated with the tumor microenvironment and stem-like cells—are still not fully elucidated. Significantly, the theory of CSCs—defined by the capacity for self-renewal, tumor development, and resistance to standard therapies—has changed the foundational views in oncology. The identification of adult stem cells in both human and murine pituitaries has encouraged the application of this model to pituitary tumors, indicating the possible presence of tumor-initiating stem-like cells even in benign neoplasms. Various studies have aimed to isolate and characterize pituitary adenoma stem-like cells (PASCs) using multiple and frequently heterogeneous experimental approaches. Even if their exact role in tumorigenesis remains to be fully elucidated, growing evidence suggests that PASCs may be involved in tumor growth and therapeutic resistance. Advancing the identification of these cells could contribute significantly to our understanding of pituitary tumor pathophysiology and open avenues for innovative diagnostic and treatment approaches [37,38].

A recent study, conducted in accordance with the WHO 2022 classification guidelines, identified a novel subtype of pituitary tumors termed plurihormonal tumors without distinct lineage differentiation (WDLD). These entities, defined based on the co-expression of multiple lineage-specific transcription factors, were assessed for stem cell-associated marker expression. The presence of SOX2, Nestin, and CD133 was evaluated across WDLD tumors, immature PIT-1 lineage tumors, and classically differentiated adenomas using immunohistochemistry and RT-qPCR. The results demonstrated a significantly elevated expression of SOX2 in WDLD tumors relative to lineage-committed counterparts, with CD133 also observed in a subset of these tumors. These findings constitute one of the first biological descriptions of WDLD tumors and suggest a potential involvement of stem-like cells in their pathogenesis [39].

In a current study involving 113 PitNETs from 109 acromegalic patients, Sox2 immunohistochemical expression was investigated in order to establish its clinical relevance. This stem cell marker was detected in both normal pituitary tissue and growth hormone-secreting tumors. Sox2 positive cells were observed in 33% of tumors at levels ≥1%, while 22% showed sparse expression (<1%) and 45% were negative. Despite this variability, Sox2 expression was not correlated with tumor size, invasiveness, proliferative activity, or hormone levels, suggesting limited prognostic value so far. Double immunostaining showed that the majority of Sox2^+^ cells were also immunopositive for annexin A1 and S100 protein exhibiting morphological characteristics of FS cells. These findings sustain the presence of a Sox2^+^ subpopulation with stem-like properties in both normal and tumorous pituitary tissue, likely representing a non-endocrine, FS cell lineage with potential stem cell functions [34].

In a study by Koyama et al., it was demonstrated that folliculo-stellate-like (FS-like) cells significantly contribute to pituitary tumorigenesis by promoting tumor growth in vivo. Using a murine model, the authors implanted GH-producing MtT/S tumor cells with and without FS-like TtT/GF cells into nude mice. The study revealed that the association of the two cell lines led to successful tumor formation and enhanced growth hormone secretion, whereas MtT/S cells alone failed to form tumors. Histological examination revealed that FS-like cells surrounded tumor nests, suggesting that their supportive microenvironment, likely through paracrine signaling, plays a critical role in facilitating neuroendocrine tumor proliferation and progression [40]. Moreover, it was mentioned that FS cells, similar to other adenohypophyseal cell types, originate from Rathke’s pouch and possess pluripotent characteristics, enabling them to perform numerous functions including endocrine regulation, angiogenesis, and modulation of immune responses. The expression of the pituitary-specific transcription factor Ptx1 in both hormone-producing progenitor cells and FSCs further supports the common developmental origin [41,42].

In addition to FS cells, blood vessels constitute a fundamental component of the TME. Our analysis revealed significant interactions between GFAP-positive cells and vascular structures. In 27 PitNETs, FS cells were predominantly located in vascular regions. Furthermore, in certain instances, strong associations were observed between FS cells and endothelial cells.

Previous studies have suggested the interplay between FS cells and blood vessels. Horiguchi et al. identified a subpopulation of adult pituitary progenitor cells expressing both S100β and SOX2, which exhibit plasticity and multipotency. These cells were isolated from the adult rat anterior pituitary using CD9 as a membrane marker. In vitro analyses demonstrated their potential to differentiate into endothelial cells through bone morphogenetic protein (BMP) signaling pathways. Additionally, it was shown that a subpopulation of CD9/S100β/SOX2-positive pituitary progenitor cells expresses the chemokine CX3CL1 and can differentiate into endothelial cells [43,44].

Given all the facts mentioned above, as well as the ability of FS cells to produce various paracrine factors such as VEGF and basic fibroblast growth factor [5], their role in blood vessel development may offer valuable insights into understanding tumor physiopathology and progression.

## 5. Conclusions

FS cells constitute a heterogenous population of cells included in pituitary TME. Their expression in PitNETs revealed a high degree of variability depending on tumor subtype. PitNETs immunopositive for more than one pituitary hormone (including associations like GH/PRL, but also unusual IHC combinations like GH/ACTH) presented FS cells in a statistically significant percentage, suggesting their involvement in the selection of pituitary tumor lineages. The impact of FS cells regarding blood vessels density and distribution may be another important aspect in understanding the mechanisms involved in tumor development and evolution. Due to their complexity, FS cells may become important allies against PitNETs, both in terms of prognosis and therapy.

## Figures and Tables

**Figure 1 cells-14-01019-f001:**
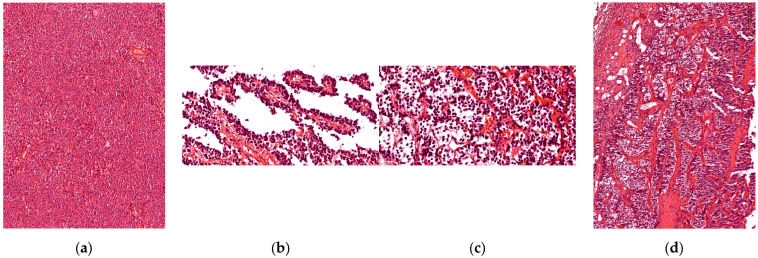
Tumoral growth pattern. (**a**) Acidophilic tumor with diffuse growth pattern; (**b**) Papillary growth pattern in longitudinal section, loose connective tissue with blood vessels in the center of the papilla; (**c**) Chromophobe tumor with alveolar growth pattern; (**d**) Trabecular growth pattern, with thick connective trabeculae inside the tumor.

**Figure 2 cells-14-01019-f002:**
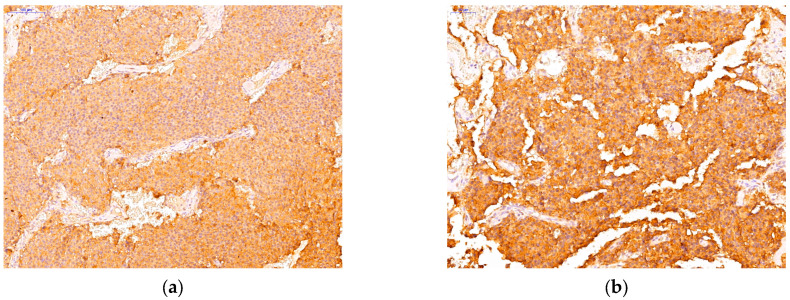
An example of an unusual plurihormonal PitNET immunopositive for GH, ACTH. (**a**) positive immunohistochemical reaction for GH; (**b**) positive immunohistochemical reaction for ACTH.

**Figure 3 cells-14-01019-f003:**
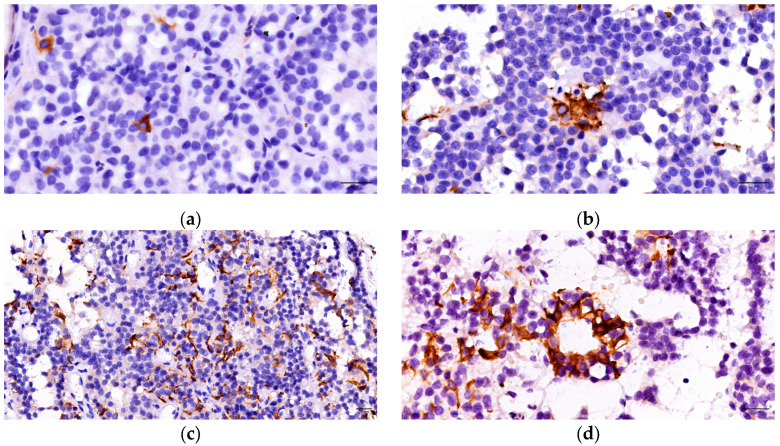
FS cells—forms of distribution: (**a**) isolated FC cell; (**b**) a group of FS cells; (**c**) diffuse network of FS cells; (**d**) FS cells forming a follicle.

**Figure 4 cells-14-01019-f004:**
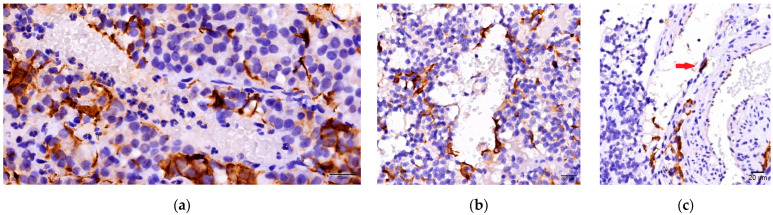
FS cells and blood vessels interactions. (**a**) FC cell grouped around blood vessels; (**b**) FC cell distributed along blood vessel wall; (**c**) Direct contact between FC cell and endothelial cell (arrow).

**Table 1 cells-14-01019-t001:** Antibodies used for anterior pituitary hormones, transcription factors and FS cells.

	Expression Pattern	Company	Clone	Dilution Factor
GH	Cytoplasmic	Dako Agilent ^1^	Polyclonal rabbit anti-human	1:400
PRL	Cytoplasmic	Dako Agilent	Polyclonal rabbit anti-human	1:300
TSH	Cytoplasmic	Thermo Fisher Scientific ^2^	TSH01 + TSH02	1:400
ACTH	Cytoplasmic	Dako Agilent	C93	1:50
FSH	Cytoplasmic	Thermo Fisher Scientific	FSH03	1:500
LH	Cytoplasmic	Thermo Fisher Scientific	LH01	1:500
PIT1	Nuclear	Thermo Fisher Scientific	Rabbit polyclonal antibody	1:500
Anti-Tpit	Nuclear	Abcam ^3^	CL6251	1:1000
SF1	Nuclear	Invitrogen (Thermo Fisher Scientific)	SF1 antibody PA5-79984	1:500
GFAP	Cytoplasmic	Leica Bond ^4^	GA5	ready to use

GH—growth hormone, PRL—prolactin, TSH—thyroid stimulating hormone, ACTH—adrenocorticotropic hormone, FSH—follicle-stimulating hormone, LH—luteinizing hormone, PIT1—pituitary-specific transcription factor, TPIT—T-box transcription factor, SF1—steroidogenic factor 1, GFAP—glial fibrillar acidic protein. ^1^ Dako Agilent-Agilent Technologies, 5301 Stevens Creek Blvd, Santa Clara, CA 95051, United States. ^2^ Thermo Fisher Scientific-Thermo Fisher Scientific, 168 Third Avenue, Waltham, MA USA 02451. ^3^ Abcam-Abcam Limited Discovery Drive, Cambridge Biomedical Campus, Cambridge, CB2 0AX, UK. ^4^ Leica Bond-Leica Biosystems, Newcastle Ltd., Newcastle Upon Tyne NE 12 8EW, UK.

**Table 2 cells-14-01019-t002:** PitNETs subtypes.

	GH	PRL	TSH	ACTH	FSH	LH	PIT1	TPIT	SF1
Somatotroph (*n* = 19)	+ (*n* = 19)	-	-	-	-	-	+ (*n* = 19)	-	-
Mammosomatotroph (*n* = 16)	+ (*n* = 16)	+ (*n* = 16)	-	-	-	-	+ (*n* = 16)	-	-
Plurihormonal PIT-1 positive (*n* = 5)	+ (*n* = 5)	+ (*n* = 4)	+ (*n* = 5)	-	-	-	+ (*n* = 5)	-	-
Corticotroph (*n* = 7)	-	-	-	+ (*n* = 7)	-	-	-	+ (*n* = 7)	-
Gonadotroph (*n* = 14)	-	-	-	-	+ (*n* = 8)	+ (*n* = 14)	-	-	+ (*n* = 14)
Unusual Plurihormonal (*n* = 11)	+ (*n* = 11)	-	+ (*n* = 1)	+ (*n* = 7)	-	+ (*n* = 4)	+ (*n* = 11)	+ (*n* = 7)	+ (*n* = 4)
Null cell (*n* = 5)	-	-	-	-	-	-	-	-	-

GH—growth hormone, PRL—prolactin, TSH—thyroid stimulating hormone, ACTH—adrenocorticotropic hormone, FSH—follicle-stimulating hormone, LH—luteinizing hormone, PIT1—pituitary-specific transcription factor, TPIT—T-box transcription factor, SF1—steroidogenic factor 1, *n*—number of cases, ”+”—immunopositive reaction, “-“—no IHC reaction.

**Table 3 cells-14-01019-t003:** Association between PitNET subtype and distribution of folliculo-stellate cells.

PitNET Subtype	Absent	Isolated	Small Groups	Diffuse Networks
	*n* (%)	*n* (%)	*n* (%)	*n* (%)
Somatotroph	10 (12.99)	1 (1.3)	4 (5.19)	4 (5.19)
Mammosomatotroph	0 (0)	5 (6.49)	6 (7.79)	5 (6.49)
Plurihormonal PIT-1 positive	0 (0)	3 (3.9)	1 (1.3)	1 (1.3)
Corticotroph	2 (2.6)	2 (2.6)	2 (2.6)	1 (1.3)
Gonadotroph	4 (5.19)	3 (3.9)	2 (2.6)	5 (6.49)
Unusual plurihormonal	2 (2.6)	2 (2.6)	3 (3.9)	4 (5.19)
Null cell	4 (5.19)	0 (0)	1 (1.3)	0 (0)
Total	22 (29.9)	16 (20.78)	19 (24.68)	20 (25.97)

## Data Availability

The original contributions presented in this study are included in the article. Further inquiries can be directed to the corresponding author.

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
