# Peer review of "An Old New Friend: Folliculo-Stellate Cells in Pituitary Neuroendocrine Tumors"

_cells, 2025, doi:10.3390/cells14131019_

Round 1
Reviewer 1 Report
Comments and Suggestions for Authors
In the present study, authors examined folliculo-stellate cells in 77 PitNETs obtained by 17 transsphenoidal surgery, using glial fibrillary acidic protein (GFAP) as an immunohistochemical 18 marker. Immunohistochemistry for anterior pituitary hormones and transcription factors was per-19 formed to accurately classify the tumors. Many thanks for the opportunity to read your manuscript.
Major comments
Introduction: The authors need to state what purpose and what kind of research was conducted in this study.
Materials and Methos: Fixing solution should be indicated. The number that passed the ethics review should be listed.
Results: Scale bars, not objective lens magnification, should be included in all figures. And figures should be shown without digital zoom. Figure 2 is not clear and should be replaced. Double and triple staining data with hormones or S100 or GFAP is needed.
Discussion: This is a repetition of what was said in the Introduction and should say more about the tumorigenesis of FS cells.
Several studies on FS cells and vascular endothelial cell differentiation have already suggested a link (Horiguchi et al. DOI:10.1038/s41598-018-23923-0 and DOI 10.1007/s00418-020-01862-0).
Minor comments
Several abbreviations not spelled out are seen, and several that are spelled out but not using abbreviations (FC cells,
Author Response
Thank you very much for taking the time to review this manuscript to such a great level of detail. Based on the findings we consider we managed to significantly increase the quality of our work. Please find the detailed responses below and the corresponding revisions highlighted in the re-submitted files.
|
Comments 1: Introduction: The authors need to state what purpose and what kind of research was conducted in this study. |
|
Response 1: Thank you for pointing this out, we have added the requested information at the end of the Introduction chapter [page 2 lines 67-69]. “The current study is a retrospective observational analysis aiming to evaluate the characteristics and distribution of FS cells within PitNETs, in relation to their specific subtypes, as defined according to current immunohistochemical classification.” |
|
Comments 2: Materials and Methos: Fixing solution should be indicated. The number that passed the ethics review should be listed. |
|
Response 2: Agree. We have, accordingly, revised the Materials and Methods to emphasize these points. The ethics committee approval number was listed on page 2 line 74 and fixing solution was listed on page 2 line 78. “The study has the approval of the University of Medicine and Pharmacy “Victor Babes” Ethics Committee (Ethics Approval no.102/03.10.2022 rev 2025).” “tissue fixation was performed using 10% neutral buffered formalin” |
|
Comments 3: Results: Scale bars, not objective lens magnification, should be included in all figures. And figures should be shown without digital zoom. Figure 2 is not clear and should be replaced. |
|
Response 3: Agree. We have, accordingly, revised these points. • In the attached microscopy folder scale was added to original slide images. Figure 2 images have been replaced with another case showing clear immunopositivity for both GH and ACTH, offering a clear cytoplasmic reaction for each hormone. • In the article body, Figure 1 has been replaced with a cropped version of the original, including the scale. • In the article body, Figure 2 images have been substituted with new cropped originals. • In the article body, Figure 3 images have been replaced by the same originals, now containing the scale. • In the article body, Figure 4 has been replaced with a cropped version of the original, including the scale. |
|
Comments 4: Results: Double and triple staining data with hormones or S100 or GFAP is needed. |
|
Response 4: Regarding double and triple immunostaining with pituitary hormones and GFAP/S100, the technical possibilities are detailed in the 'Materials and Methods' section. The immunohistochemistry equipment currently available does not support double or triple staining involving antibodies that share the same subcellular localization—in this case, cytoplasmic. Practically, such staining would result in overlapping cytoplasmic signals, which would be visually inconclusive due to the lack of contrast. Moreover, an additional limitation arises from the fact that both the antibodies used for pituitary hormones and the one targeting GFAP are anti-human, and thus derived from the same species. This would lead to cross-reactivity between the primary antibodies, further compromising the specificity and interpretability of the staining. |
|
Comments 5: This is a repetition of what was said in the Introduction and should say more about the tumorigenesis of FS cells. Several studies on FS cells and vascular endothelial cell differentiation have already suggested a link (Horiguchi et al. DOI:10.1038/s41598-018-23923-0 and DOI 10.1007/s00418-020-01862-0).
|
|
Response 5: We sincerely thank you for highlighting such valuable and insightful references, which have significantly contributed to improving the quality of our manuscript. After carefully reading the 2 articles we have decided to link our study in the context of the results presented there. Additionally, we have integrated three additional references, one that covers in depth the process of tumorigenesis and two other that focus on the complexity of FS cells [page 8 lines 266-291]. “In a study by Koyama et al. it was demonstrated that folliculo-stellate-like (FS-like) cells significantly contribute to pituitary tumorigenesis by promoting tumor growth in vivo. Using a murine model, the authors co-implanted GH-producing MtT/S tumor cells with FS-like TtT/GF cells into nude mice, observing that only the combination led to successful tumor formation and enhanced growth hormone secretion, whereas MtT/S cells alone failed to form tumors. Histological examination revealed that FS-like cells surrounded tumor nests, suggesting that their supportive microenvironment, likely through paracrine signaling, plays a critical role in facilitating neuroendocrine tumor proliferation and progression [38]. Moreover, it was mentioned that FS cells, similar to other adenohypophyseal cell types originate from Rathke’s pouch and possess pluripo-tent characteristics, enabling them to perform numerous functions including endocrine regulation, angiogenesis, and modulation of immune responses. The expression of the pituitary-specific transcription factor Ptx1 in both hormone-producing progenitor cells and FSCs further supports the common developmental origin [39,40]. In addition to FS cells, blood vessels constitute a fundamental component of the TME. Our analysis revealed significant interactions between GFAP positive cells and vascular structures. In 27 PitNETs, FS cells were predominantly located in vascular re-gions. Furthermore, in certain instances, strong associations were observed between FS cells and endothelial cells. Previous studies have suggested the interplay between FS cells and blood vessels. Horiguchi et al. identified a subpopulation of adult pituitary progenitor cells expressing both S100β and SOX2, which exhibit plasticity and multipotency. These cells were iso-lated from the adult rat anterior pituitary using CD9 as a membrane marker. In vitro analyses demonstrated their potential to differentiate into endothelial cells through bone morphogenetic protein (BMP) signaling pathways. Additionally, it was shown that a subpopulation of CD9/S100β/SOX2-positive pituitary progenitor cells express the chemokine CX3CL1 and can differentiate into endothelial cells [41, 42]. Given all the facts mentioned above, as well as the ability of FS cells to produce various paracrine factors such as VEGF and basic fibroblast growth factor [5], their role in blood vessel development may offer valuable insights into understanding tumor physiopathology and progression.” |
|
4. Response to Comments on the Quality of English Language |
|
Point 1: Several abbreviations not spelled out are seen, and several that are spelled out but not using abbreviations (FC cells) |
|
Response 1: Thank you for pointing this out. We have revised the abbreviations by removing the FC cells and bFGF abbreviations from the text as they were not necessary and have replaced in all article parts the following: Folliculo-stellate cells -> FS cells Tumoral microenvironment -> TME Pituitary neuroendocrine tumors -> PitNETs Glial fibrillary acidic protein -> GFAP Vascular endothelial growth factor -> VEGF |
Reviewer 2 Report
Comments and Suggestions for Authors
The role of FS cells in pituitary neuroendocrine tumors is not well understood and the current manuscript adds to the understanding that the relationship between phenotypically diverse FS cells and tumor cells is complex with good immunohistochemical evidence that there may be relationships between tumor subtypes and the profiles of factor production by FS cells. To confirm the results laser capture microdissection and gene expression analysis would be ideal, as there can be a concern that products identified by IHC may not have been produced by the apparently positive cell. Given the limitations of IHC there is only one concern with the current presentation and that is that not all FC express GFAP, which was used here for their identification. In fact, GFAP is currently thought to be a marker of FC early in their development (eg. Horvath, Eva; Kovacs, Kalman (2002-01-01). "Folliculo-stellate Cells of the Human Pituitary: A Type of Adult Stem Cell?". Ultrastructural Pathology. 26 (4): 219–228.) This possibility and the impact on the current studies should be discussed.
Author Response
Thank you for your time, your positive feedback, and the highly valuable observations you have provided. Based on your suggestions, we have included two relevant references in the Discussion section, which we believe contribute to improving the overall quality of the manuscript. In this study, our aim was to highlight the distribution of these cells according to the subtypes of pituitary tumors, particularly given the lack of targeted studies addressing this specific aspect. Indeed, folliculo-stellate cells represent a highly complex population, encompassing multiple subtypes—some of which remain only partially characterized or entirely unexplored in the current literature [page 8 lines 274-279]
"Moreover, it was mentioned that FS cells, similar to other adenohypophyseal cell types originate from Rathke’s pouch and possess pluripo-tent characteristics, enabling them to perform numerous functions including endocrine regulation, angiogenesis, and modulation of immune responses. The expression of the pituitary-specific transcription factor Ptx1 in both hormone-producing progenitor cells and FSCs further supports the common developmental origin [39,40]."
Round 2
Reviewer 1 Report
Comments and Suggestions for Authors
The areas this reviewer pointed out have been appropriately corrected.